# Diabetes with COVID-19 was a significant risk factor for mortality, mechanical ventilation, and renal replacement therapies: A multicenter retrospective study in Japan

Hirotsugu Suwanai[1]*, Masato Kanda[2,3], Kazuharu Harada[4], Keitaro Ishii[1], Hajime Iwasaki[1], Natsuko Hara[1], Yoshio Kobayashi[2], Hajime Matsumura[5], Takahiro Inoue[3], Ryo Suzuki[1]

1 Department of Diabetes, Metabolism, and Endocrinology, Tokyo Medical University, Tokyo, Japan,
2 Department of Cardiovascular Medicine, Graduate School of Medicine, Chiba University, Chiba, Japan,
3 Healthcare Management Research Center, Chiba University Hospital, Chiba, Japan, 4 Department of Health Data Science, Tokyo Medical University, Tokyo, Japan, 5 Department of Plastic and Reconstructive Surgery, Tokyo Medical University, Tokyo, Japan

* suwanai-h@umin.ac.jp

## Abstract

We conducted a multicenter retrospective cohort study across 38 hospitals in Chiba, Japan, between February 1, 2020 and November 31, 2021 to investigate the effect of coronavirus disease 2019 (COVID-19) on patients with diabetes mellitus receiving inpatient care. We collected inpatient medical data through Diagnosis procedure combination (DPC), the diagnoses and payment system of medical insurance, from each hospital. We excluded patients younger than 18 years, those who were pregnant, and those who had diabetes but were not treated with diabetic medication. A total of 10,776 patients were included: 7,679 in the non-diabetic (control) group and 3,097 in the diabetic group. Patients in the diabetic group were older and had a higher body mass index (BMI) than those in the control group. In the diabetes group, 88.4% of the patients were treated with insulin therapy and 44.2% were treated with oral hypoglycemic agents. The length of hospital days was significantly longer in the diabetes group. The in-hospital mortality rate was significantly higher especially between 50 and 59 years old. The rates of in-hospital mortality, mechanical ventilation, intensive care unit (ICU) admission, renal replacement therapies such as hemodialysis (HD), and continuous hemodiafiltration (CHDF) were all higher, even after adjusting for age, sex, BMI, and ambulance use. In conclusion, diabetes was a significant risk factor of the severe clinical outcomes especially for in-hospital mortality, mechanical ventilation usage, ICU admission, HD, and CHDF in Japan.

## Introduction

The severe acute respiratory syndrome coronavirus-2 (SARS-CoV-2) is highly contagious and has caused the coronavirus disease 2019 (COVID-19) pandemic. As the COVID-19 continued to spread, researchers have investigated the factors contributing to the severity of the illness

**Data availability statement:** Data cannot be shared publicly because of the research protocol approved by the ethics committee of Chiba University Hospital (approval number 3309 since 27th June 2022). Data are available from the Chiba University Hospital for researchers who meet the criteria for access to confidential data.

**Funding:** The author(s) received no specific funding for this work.

**Competing interests:** The authors have declared that no competing interests exist.

and its treatment, driven by its high mortality rate and the prevalence of cases. Hyperglycemia and diabetes were independent risk factors during the SARS-COV-1 outbreak in 2002–2003 [1]. Additionally, diabetes was reported to be a risk for severe disease in 2012 Middle East Respiratory Syndrome Coronavirus (MERS-CoV) [2]. Moreover, not only diabetes but also older age, male, hypertension, obesity, cancer, and kidney failure were identified as the multiple risk factors of COVID-19 [3–5].

COVID-19 is associated with multi-organ damage, making the clinical manifestations critical for understanding the pathophysiology of the disease. It often leads to acute respiratory distress syndrome (ARDS), acute kidney injury (AKI) requiring hemodialysis, and other forms of organ failures [6]. These complications contribute to increased admissions to intensive care unit (ICU), where the use of mechanical ventilation and dialysis is common [7]. Treatment of COVID-19 in Japan followed Japanese guidelines, including the introduction of mechanical ventilation [8]. Although the mortality rate from SARS-CoV-2 infections has been relatively low in Japan, an elevated risk of mortality has been reported in patients with diabetes [9]. However, comprehensive studies examining the treatment and outcomes of COVID-19 in patients with diabetes remain limited.

Here, we conducted a study using real-world data on inpatient medical information and Diagnosis Procedure Combination (DPC) data in Japan to analyze the background of hospitalized patients. The aims of this research were to assess the impact of COVID-19 on patients with diabetes over time and to investigate their association with mechanical ventilation, ICU admissions, hemodialysis, and continuous hemodiafiltration (CHDF), as well as the implications for medical resource utilization. Finally, we evaluated the various effects of COVID-19 on mortality among patients with diabetes. The findings of this study may offer valuable insights for managing future infectious disease outbreaks.

## Materials and methods

### Data source

The DPC system is a classification system for hospital reimbursements from insurers with regular inspections by government in Japan. The DPC database contains inpatient clinical information, including disease diagnoses, comorbidities, medications, and medical treatments such as mechanical ventilation, intensive care units (ICU), and hemodialysis. Diseases were indicated according to the International Classification of Diseases, Tenth Revision (ICD-19) codes [10].

### Study population

The fully anonymized DPC data of 11,601 patients diagnosed and hospitalized with COVID-19 between February 1, 2020 and November 31, 2021, were extracted from 38 hospitals. The Data were accessed for research purposes and obtained on 6th July 2022. These hospitals had a research collaboration with the Chiba University Hospital Director's Planning Office and maintained the DPC data. This retrospective study was conducted using these data. The data were fully anonymized, and authors did not have access to information that could identify individual participants during or after data collection.

### Outcomes and statistical analysis

The primary outcome was in-hospital mortality. Parameters of mechanical ventilation, ICU, hemodialysis, and continuous hemodiafiltration were investigated. All statistical analyses were performed using SPSS Statistics 28 (IBM, Armonk, NY, USA). Continuous variables were

assessed using Student's t-test, and categorical variables were analyzed using Fisher's exact test. Logistic regression analyses were performed to determine the association between diabetes and outcomes after adjusting for confounders, including age, sex, body mass index (BMI), and ambulance. A P < 0.05 was considered statistically significant.

## Ethics statements

This study adhered to the ethical standards of the Declaration of Helsinki. The study was approved by the ethics committee of Chiba University Hospital (approval number 3309 since 27th June 2022). This research is not registered on the clinical trials registry system because the design of this study is retrospective and public by opt-out method presenting on hospital websites. The requirement for direct informed consent was waived owing to the anonymity of the data. We believe that the clinical trials registry system would not affect the results and conclusions of the study strongly.

## Result

Of the 11,601 cases, 825 were excluded:402 were under the age of 18, 114 were pregnant, and 309 were cases with diabetes mellitus but not on diabetic medication. Although diagnosis of diabetes using DPC data has been reported to be accurate [11], we excluded cases of diabetes without medication, which were less than 3% among all cases, to analyze diabetes group more accurately. Of the remaining 10,776 cases, 7,679 and 3,097 were classified in the non-diabetic (control) and diabetic group, respectively (Fig 1). Among 3,097 cases of diabetes, those are mostly type 2 diabetes except 15 cases were type 1 diabetes (0.48%).

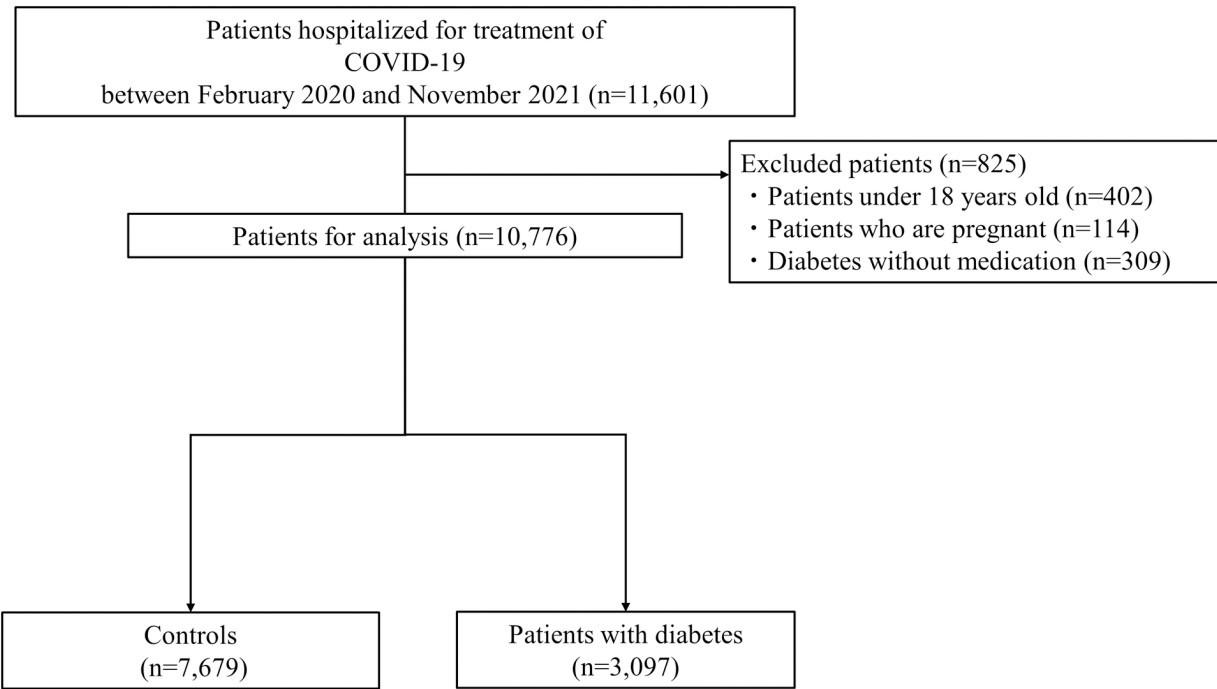

**Fig 1. Flow chart of the population selection for this study.** The data was extracted between February2020 and November 2021. Among 11,601 cases, 825 were excluded because 402 were under the age of 18, 114 were pregnant, and 309 were cases with diabetes mellitus but not on diabetic medication. The group of control had 7,679 cases while the group of diabetes had 3,097 cases.

The background results showed a correlation of age, male sex, and body mass index (BMI). The mean age of patients in the diabetic group was 67.4 years (55.7 years in the control group, p-value < 0.001). The proportion of males was higher (68.2%) in the diabetes group than in the control group (54.9%) (odds ratio [OR] 1.75, 95% confidence interval [CI] 1.60–1.91, p-value < 0.001). For patients with diabetes, the BMI was 25.6 kg/m², whereas that of the control was 23.6 kg/m² (p < 0.001). The existence of diabetes augmented the probability of hospitalization via ambulance (22.2% in the control group vs 40.7% in the diabetes group, OR 2.41, 95% CI 2.21–2.63, p < 0.001). Regarding diabetes treatment, oral hypoglycemic agents were administered in 44.2% of patients with diabetes. The rate of insulin use was 88.4%, indicating a high rate of insulin therapy in the diabetes group (Table 1). Among hypoglycemic agents, DPP-4 inhibitors were the most common medication (42.4%). All background of hypoglycemic agents is presented in data (S1 Table).

The therapeutic interventions for patients with diabetes were more severe than those for controls. The rate of steroid use was significantly higher in the diabetes (77.8%) than in control (28.35) group because patients with diabetes often have pneumonia and are treated with steroid (OR 8.86, 95% CI 8.03–9.77, p < 0.001). The continuous hemodiafiltration (CHDF) and hemodialysis (HD) were introduced at a high rate in the diabetes group, indicating that they had potentially severe circulation dynamics, renal, and heart failures (CHDF: OR 14.05, 95% CI 8.49–23.26, p < 0.001; HD: OR 3.91, 95% CI 2.85–5.36, p < 0.001). Because patients with severe symptoms of COVID-19 were treated with mechanical ventilation in ICU, the usage of mechanical ventilation and ICU were more common for patients with diabetes (mechanical ventilation: OR 13.35, 95% CI 10.93–16.30, p < 0.001; ICU: OR 7.07, 95% CI 6.00–8.33, p < 0.001) (Table 2).

Patients with diabetes were hospitalized for longer periods of time. The mean length of hospital stay was 17.8 days (10.2 days in the control group), and the in-hospital mortality rate was high at 12.9% (3.5% in the control group (OR, 4.05; p < 0.001). Additionally, the terms in hospitals until mortality were longer. The mean number of days of hospitalization until mortality in the diabetes group was 19.4 days (13.4 days in the control group) (Table 2).

The number of patients varied over time. The first case of COVID-19 was confirmed in Japan on 15th January, 2020. Thereafter, the number of affected patients has increased. From 7th April to 25th May, 2020, the government issued its first emergency declaration, and the activities were restrained. The number of patients increased in the latter half of 2020. The

**Table 1. Clinical characteristics of controls and patients with diabetes. Abbreviations: BMI=body mass index, OHA=Oral hypoglycemic agents.**

|  | Total (n = 10776) | Controls (n = 7679) | Patients with diabetes (n = 3097) | p-value | Odds ratio | 95% confidence interval |
|---|---|---|---|---|---|---|
| Age (years), mean (SD) | 59.0 (20.1) | 55.7 (21.1) | 67.1 (14.5) | <0.001 |  |  |
| Male, n (%) | 6325 (58.7) | 4214 (54.9) | 2111 (68.2) | <0.001 | 1.75 | 1.60-1.91 |
| BMI (kg/m²), mean (SD) | 24.2 (5.1) | 23.6 (4.9) | 25.6 (5.4) | <0.001 |  |  |
| Ambulance, n (%) | 2963 (27.5) | 1702 (22.2) | 1261 (40.7) | <0.001 | 2.41 | 2.21-2.63 |
| OHA, n (%) | 1370 (12.7) | 0 (0.0) | 1370 (44.2) |  |  |  |
| Insulin therapy, n (%) | 2739 (25.4) | 0 (0.0) | 2739 (88.4) |  |  |  |

**Table 2. Clinical information of inpatient care for controls and patients with diabetes. Abbreviations: CHDF=continuous hemodiafiltration, ICU=intensive care unit.**

| | Total (n = 10776) | Controls (n = 7679) | Patients with diabetes (n = 3097) | p-value | Odds ratio | 95% confidence interval |
|---|---|---|---|---|---|---|
| Steroid, n (%) | 4585 (42.6) | 2176 (28.3) | 2409 (77.8) | <0.001 | 8.86 | 8.03-9.77 |
| CHDF, n (%) | 117 (1.1) | 18 (0.2) | 99 (3.2) | <0.001 | 14.05 | 8.49-23.26 |
| Hemodyalysis, n (%) | 165 (1.5) | 65 (0.8) | 100 (3.2) | <0.001 | 3.91 | 2.85-5.36 |
| Mechanical ventilation, n (%) | 676 (6.3) | 123 (7.8) | 553 (17.9) | <0.001 | 13.35 | 10.93-16.30 |
| ICU, n (%) | 742 (6.9) | 216 (2.8) | 526 (17.0) | <0.001 | 7.07 | 6.00-8.33 |
| Days in hospitals, mean (SD) | 12.4 (11.4) | 10.2 (8.5) | 17.8 (15.3) | <0.001 | | |
| In-hospital mortality, n (%) | 671 (6.3) | 271 (3.5) | 400 (12.9) | <0.001 | 4.05 | 3.45-4.78 |
| Days in hospitals until mortality, mean (SD) | 17.0 (14.6) | 13.4 (11.5) | 19.4 (15.9) | <0.001 | | |

number of patients decreased due to the issuance of a second emergency declaration from 8[th] January to 21[st] March in2021 (Fig 2A).

Vaccination began from 17[th] February, 2021 starting with healthcare workers and older adults. Subsequently, the number of infected patients aged > 70 years has decreased. A third state of emergency was declared from 25[th] April to 20[th] June, 2021. The fourth emergency declaration was from 12[th] July to 30[th] September, 2021 (Fig 2B).

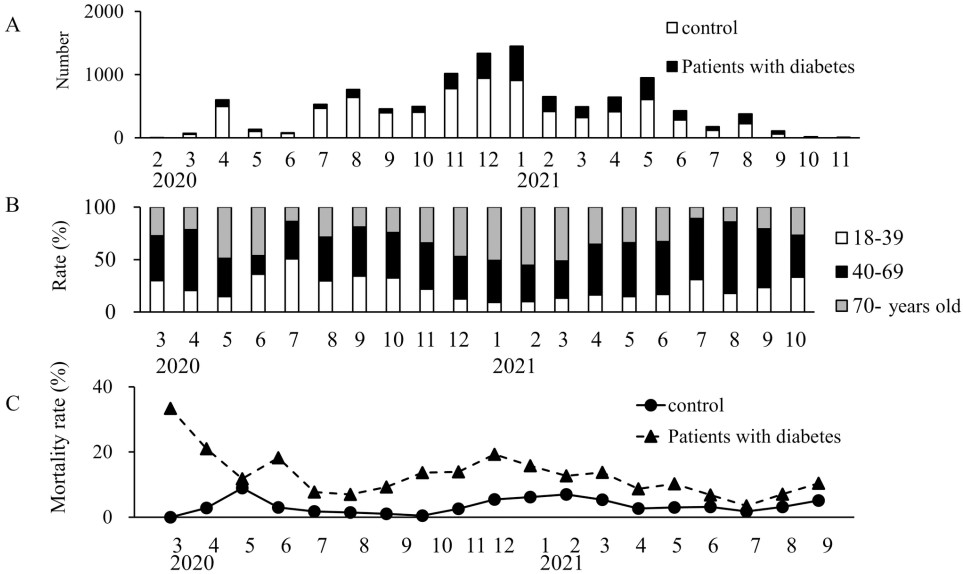

**Fig 2. Overview of control group and patients with diabetes in each month.** A) Monthly movement of number of patients of each group B) Monthly rate of patient age. The graph shows each group aged between 18 and 39 years; 40 and 69 years; and over 70 years C) Monthly mortality in the control group and patients with diabetes.

Mortality remained higher in the diabetes group than in the control group. Mortality was relatively high among patients with diabetes from March to June 2020, during the early stages of the COVID-19 pandemic. With an increase in the number of hospitalized patients, there was an increasing trend in mortality from October to December 2020. Thereafter, mortality declined as the number of hospitalizations decreased and vaccines became more widely available in 2021. After December 2021, omicron strains, which were reported to have a lower mortality rate, became predominant, and the data were excluded from the analysis (Fig 2C).

We conducted a binary logistic regression analysis to evaluate the contribution of various factors to mortality of COVID-19. The analysis included age, BMI, sex (male), diabetes, and ambulance use as response variables. Among these, diabetes emerged as the most significant predictor of mortality (OR 2.67, p value < 0.01) (S2 Table).

Subgroup analyses of the age were performed to determine the correlation between diabetes and mortality. We observed an increase in diabetes prevalence with age, peaking between 70 and 79 years old. The prevalence of diabetes was higher in male patients than in female (Fig 3A). The OR was high in patients aged > 40 years. In particular, the peak OR was 12.8 (95%CI 3.71–44.1, p < 0.01) in ages between 50 and 59 years, indicating that diabetes was a strong risk factor for COVID-19 mortality in the middle-aged generation (Fig 3B and 3C).

Regression analysis was performed to test the effect of diabetes on the outcomes. The ORs with diabetes for in-hospital mortality, mechanical ventilation, ICU, HD, and CHDF were significantly higher compared to the control group in unadjusted analysis (in-hospital mortality: OR 4.05, 95%CI 3.45–4.76; mechanical ventilation: OR 13.35, 95%CI 10.93–16.32; ICU:7.07, 95%CI 6.00–8.33; HD: OR 3.91 95%CI 2.85-5.36; and CHDF: OR 14.06, 95%CI 8.49–23.27). Because there were significant differences in age, sex, BMI, and ambulance use between the

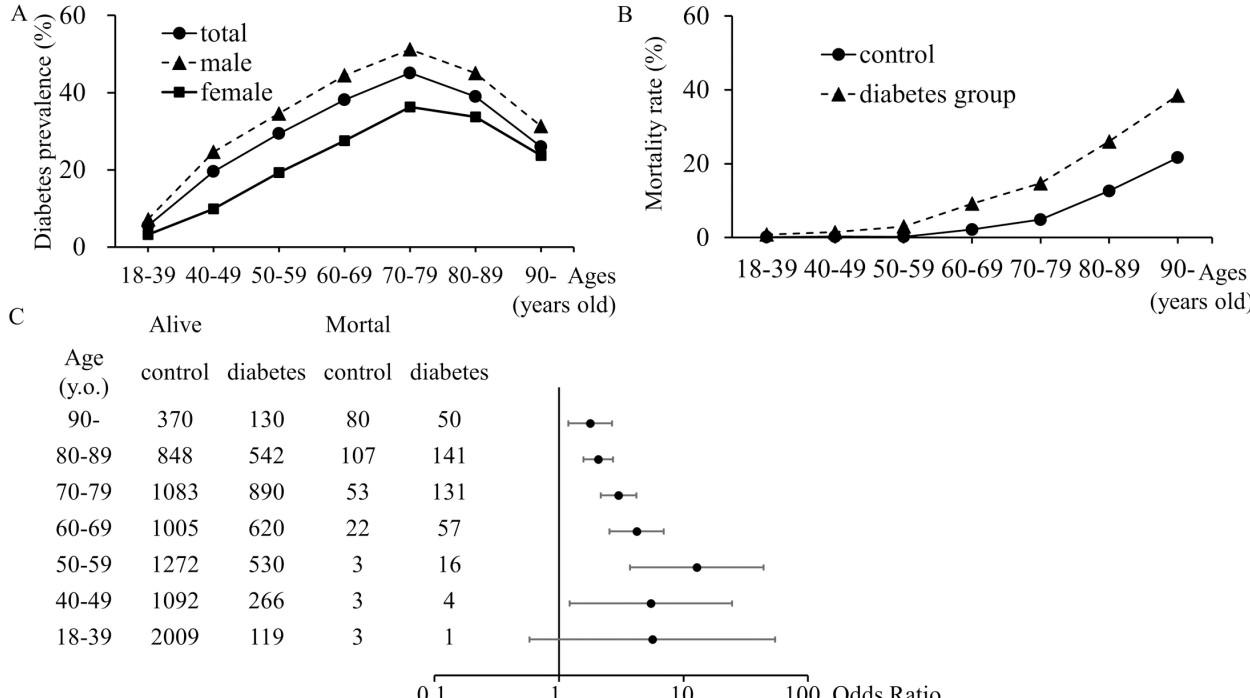

**Fig 3. Subgroup analyses of the age in diabetes prevalence and mortality.** A) Diabetes prevalence classified according to age and sex. B) Mortality rate of control and patients with diabetes in each generation. C) Odds ratio of in-hospital mortality for patients with diabetes compared to controls at each generation.

control and diabetes groups, adjustment model of multiple regression analysis was conducted. As we conducted multiple regression analysis using adjustment with age, sex, BMI, ambulance, the ORs for all outcomes remained significant, proving that diabetes contributed as an independent factor of outcomes (in-hospital mortality: OR 2.67, 95%CI 2.17–3.28; mechanical ventilation: OR 8.31, 95%CI 6.61–10.45; ICU:4.19, 95%CI 3.46–5.08, HD: OR 3.95 95%CI 2.69–5.79; and CHDF: OR 10.40, 95%CI 5.70–18.99) (Fig 4). All those findings indicated that diabetes was a significant risk factor of the severe clinical outcomes especially for in-hospital mortality, mechanical ventilation use, ICU admission, renal replacement therapies such as HD, and CHDF.

## Discussion

In this study, we analyzed the effects of COVID-19 on patients with and without diabetes in Japan using real-world data. Diabetes was a significant factor in in-hospital mortality, mechanical ventilation, ICU stay, HD, and CHDF, even after adjusting for age, sex, BMI, and ambulance use, indicating that diabetes is associated with a worse prognosis for COVID-19. In addition, the mortality rate was higher in the diabetes group, especially among those aged 18–79 years, than in the control group, indicating that the presence of diabetes increases risk in the younger generation.

Previous studies have heavily implicated diabetes and COVID-19 mortality rates [12,13]. Our data show a similar trend. Additionally, we analyzed treatment data and found that diabetes itself was an independent risk factor for mortality, mechanical ventilation, ICU stay, HD, and CHDF, even after adjusting for age, sex, BMI, and ambulance use.

The strength of our study is that we used real-world data and based on actual treatments administered during hospitalization. We believe that it is particularly important to comprehensively analyze mortality rates and interventions that require diabetes treatment, classified

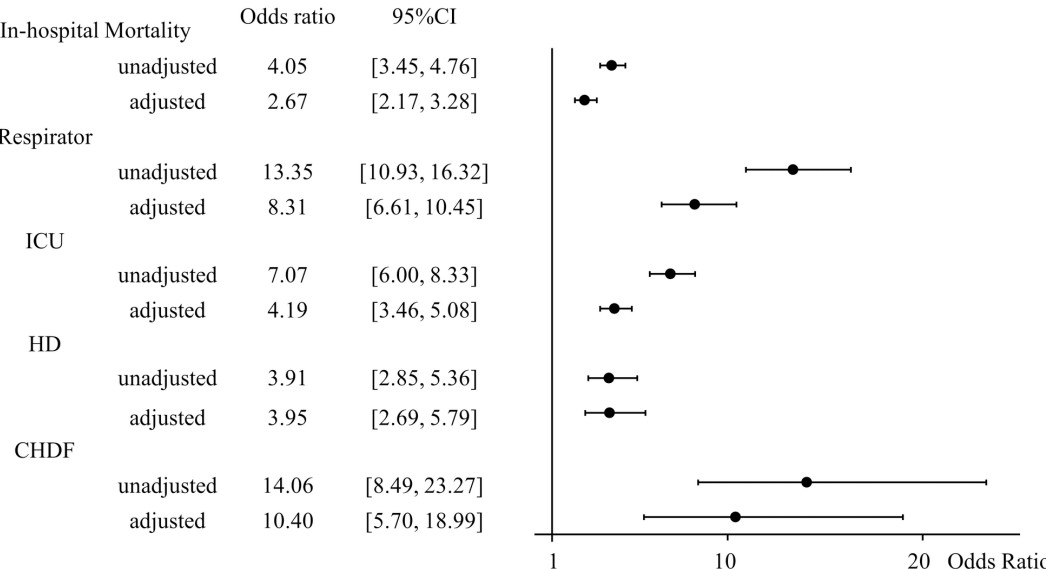

**Fig 4. Forest plot of multiple regression analysis for in-hospital mortality, mechanical ventilation, ICU, HD, and CHDF.** Regression analysis was performed to test the effect of diabetes on the outcomes. The ORs with diabetes for in-hospital mortality, mechanical ventilation, ICU, HD, and CHDF were significantly higher compared to the control group in unadjusted analysis. Multiple regression model with adjustment of age, sex, BMI, and ambulance use was used for each outcome.

into age group. The results of this study revealed that the risk of mortality was significantly higher in the group that required diabetes treatment.

It is believed that COVID-19 infects not only the lungs but also kidneys and pancreatic beta cells, causing inflammation [14,15]. The viral infection depends on the expression of angiotensin-converting enzyme 2 (ACE2) and TMPRSS-2 transmembrane protease, which are the main cellular factors involved in viral entry. ACE2 is expressed in the lungs, and the virus infects the lungs and causes ARDS [16]. COVID-19 stimulates strong immune response, resulting in a cytokine storm. It is possible that patients with diabetes are unable to suppress inflammation and that the virus enters the organs via those proteins, further exacerbating the inflammation and hyperglycemia [15,17].

In this study, diabetes was a significant factor in in-hospital mortality, mechanical ventilation, ICU stay, HD, and CHDF. AKI associated with COVID-19 is mainly caused by proximal tubular dysfunction. ACE2 expression is abundant in the proximal tubules of the kidney, and it is believed that SARS-CoV-2 infects these cells, causing damage [18,19]. AKI complications were reported in 36.6% of patients admitted with COVID-19 [20]. The combination of renal and respiratory failure is thought to have led to the use of mechanical ventilation, continuous hemodialysis (CHDF), and hemodialysis (HD) to support critically ill patients in the ICU.

SARS-CoV-2 infects pancreatic beta cells and reduces their function. In this study, the diabetes group was defined as patients receiving diabetes care during hospitalization. It is assumed that diabetes treatment was performed because of elevated blood glucose levels. This indicates that the virus may have infected pancreatic beta cells, potentially leading to a significant impact on blood glucose levels [21–23].

Japanese individuals are characterized by lower insulin secretion than Westerners. Asians, including the Japanese, often develop type 2 diabetes, even if their obesity is relatively modest compared to Westerners [24,25]. Due to the government's emergency declaration, activities were restricted. The following factors were noted among those who gained weight after the spread of COVID-19: shortened sleeping hours during long periods of time indoors, increased snacking after dinner, lack of exercise, overeating due to stress, and loss of motivation to restrict eating, suggesting the importance of daily lifestyle habits [26]. As obese individuals with high insulin resistance may easily become hyperglycemic due to infection, it is necessary to raise awareness of the importance of proper weight control in preventing the spread of infection.

This study had some limitations. Our data provide information on diagnoses and treatments during hospitalization but not prior to hospitalization. Additionally, our data did not include information on blood examinations, vital signs, laboratory parameters, radiographic data including PaO2/FiO2 ratio and severity classification of COVID-19. We also could not obtain information on vaccination histories. However, we believe that our study sheds light on the clinical characteristics of COVID-19 and diabetes for improved medication in the future.

In conclusion, diabetes was an independent risk factor for in-hospital mortality, mechanical ventilation, ICU stay, HD, and CHDF, even after adjusting for age, sex, BMI, and ambulance use. The mortality rate was higher in the diabetes group, especially among those aged 18–79 years, than in the control group. Diabetes has been confirmed as a risk factor for clinical severity and in-hospital mortality with COVID-19.

## Supporting information

**S1 Table. Content of hypoglycemic agents for patients with diabetes.** Abbreviations: DPP-4 = dipeptidyl peptidase 4, SGLT2 = sodium glucose co-transporter 2, AGI=Alpha-glucosidase inhibitor.
(DOCX)

**S2 Table. Multiple regression analysis of mortality adjusted with each factor, diabetes, age, sex (male), BMI, and ambulance.** Abbreviation: BMI=Body mass index.
(DOCX)

**S1 File. Spreadsheet of hypoglycemic agents with generic names for patients with diabetes.**
(CSV)

## Acknowledgments

The authors are grateful to all participants in this study. We would like to thank Editage (www.editage.jp) for English language editing.

## Author contributions

**Conceptualization:** Hirotsugu Suwanai.

**Data curation:** Masato Kanda, Kazuharu Harada, Keitaro Ishii, Hajime Iwasaki, Natsuko Hara, Takahiro Inoue.

**Formal analysis:** Keitaro Ishii, Hajime Iwasaki, Natsuko Hara.

**Supervision:** Hajime Matsumura, Takahiro Inoue, Ryo Suzuki.

**Writing – original draft:** Hirotsugu Suwanai.

**Writing – review & editing:** Natsuko Hara, Yoshio Kobayashi, Hajime Matsumura, Takahiro Inoue, Ryo Suzuki.

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
