## [Decision Letter · Decision Letter 0]

20 Oct 2024

PONE-D-24-22312Diabetes with COVID-19 was a significant risk factor for mortality, respirator use, and renal replacement therapies: A multicenter retrospective study in JapanPLOS ONE

Dear Dr. Suwanai,

Thank you for submitting your manuscript to PLOS ONE. After careful consideration, we feel that it has merit but does not fully meet PLOS ONE’s publication criteria as it currently stands. Therefore, we invite you to submit a revised version of the manuscript that addresses the points raised during the review process.

The reviewers have recommended publication, but also suggest significant revisions to your manuscript.  Therefore, I invite you to respond to the reviewers' comments and revise your manuscript.

We look forward to receiving your revised manuscript.

Kind regards,

Fumihiro Yamaguchi

Academic Editor

PLOS ONE

Journal Requirements:

Reviewers' comments:

Reviewer's Responses to Questions

**Comments to the Author**

1. Is the manuscript technically sound, and do the data support the conclusions?

Reviewer #1: Yes

Reviewer #2: No

Reviewer #3: Partly

2. Has the statistical analysis been performed appropriately and rigorously? 

Reviewer #1: Yes

Reviewer #2: No

Reviewer #3: Yes

3. Have the authors made all data underlying the findings in their manuscript fully available?

Reviewer #1: Yes

Reviewer #2: Yes

Reviewer #3: Yes

4. Is the manuscript presented in an intelligible fashion and written in standard English?

Reviewer #1: Yes

Reviewer #2: Yes

Reviewer #3: Yes

5. Review Comments to the Author

Reviewer #1: Dear Authors

The current manuscript is outstanding. Even so, there are some questions.

The current manuscript needs to improve some points. The results are descriptive and straightforward comparisons between the two groups (diabetes mellitus and control).

Introduction

The authors should briefly comment on the relationship between hypertension and COVID-19 and briefly describe the association between COVID-19 and outcomes such as AKI, the need for mechanical ventilation, and mortality.

Materials and Methods

What does it mean? The

-Diseases were indicated according to the International Classification of 83 Diseases, Tenth Revision (ICD-19) codes (6). The (????)-

Study Population

How did you ensure that you evaluated patients diagnosed with diabetes mellitus? And about COVID-19? Why didn't you run some blood tests? Was there blood glucose in the patients or glycated hemoglobin? What about creatinine, arterial blood gas analysis, and hemoglobin concentration?

Why didn't you perform some matching between the diabetes and control groups, such as age, sex, or comorbidities? If you had, it would have made comparisons more robust.

Results

Would the authors be able to inform the frequency of patients with type 1 diabetes in the sample?

It would be interesting for you to build a binary logistic regression table using mortality as the response variable.

You need to correct this sentence:

The data was extracted between February220 and November 2021

The proportion of males was higher (54.9%) in the control group than in the diabetes 132 group (68.2%) (odds ratio [OR] 1.75, 95% confidence interval [CI] 1.60–1.91). Report me about that, is 54.9% higher than 68,2%?

For patients with 133 diabetes, the BMI was 25.6%, whereas that of the control was 23.6 (p <0.001). What does it mean? BMI, 25.6 kg/m²? And 23.6 kg/m²?

What kind of hypoglycemic agents were used for patients with diabetes mellitus?

Could the authors report the PaO2/FiO2 ratio at baseline in patients undergoing mechanical ventilation?

Discussion

The authors need to discuss the hyperglycemia observed in patients with infection or even the fact that infections, including SARS-CoV-2, decompensate diabetes mellitus.

The authors need to discuss the relationship between the need for mechanical ventilation, AKI, dialysis requirement, and mortality in patients with diabetes mellitus and COVID-19.

Reviewer #2: I have some comments and questions for the authors

1. What is the reason for the existence of diabetes to augmented the probability of hospitalization via ambulance?

2. 77.8% of patients with diabetes used steroids for the treatment of pneumonia. I think it is important to know what percentage of pneumonias. What percentage were severe pneumonias?

3. 7.8% of the control group and 17.9% of the group with diabetes required mechanical ventilation. What was the main reason why the patients required this support? Do you have respiratory data such as paO2/fiO2

ratio, or others?

4. An important conclusion is that diabetes was an independent risk factor for in-hospital mortality. However, other conclusions such as risk factor for respirator use require further discussion.

5. The manuscript does not describe severity scores for the control group and the diabetes group. I think it is important to establish the severity of each group

6. We cannot see in the manuscript hemodynamic and inflammatory variables and other comorbidities of each group. As in point 5, it is important to know these data

Reviewer #3: The submitted manuscript presents an interesting topic, aiming to examine the effect of COVID-19 on patients with diabetes mellitus receiving inpatient care.

To enhance the reliability and comprehensibility of this work for researchers, the following recommendations may be considered:

1. In the introduction, line 70, you mentioned that this study was conducted using real-world data on inpatient medical information and Diagnosis Procedure Combination (DPC) data in Japan to analyze the background of hospitalized patients. However, the authors did not mention the aims of the study. I think you should clearly explain the main aims of this study.

2. The numbers and percentages in Tables 1 and 2 are inconsistent. For example, in Table 1, for the control group, the mean age in (years) is in the first line, the SD is in the second line. However, for the patient group, both are in the same line. Could you please make the numbers and percentages consistent for both Tables 1 and 2?

3. The authors concluded that diabetes was a significant risk factor for severe clinical outcomes, especially for in-hospital mortality, respirator use, ICU admission, renal replacement therapies such as HD and CHDF. The manuscript would benefit from deeper biological interpretations and a more thorough exploration of the underlying mechanisms. For example, what are the broader implications of COVID-19 on these parameters (in-hospital mortality, respirator use, ICU admission, renal replacement therapies such as HD and CHDF)? How might these findings tie into the pathophysiology of COVID-19 cases?

4. Another important point is whether the COVID-19 cases included in this study were mild, moderate, or severe. If possible, could you identify the effect of COVID-19 on mortality, respirator use, ICU admission, and renal replacement therapies in terms of COVID-19 severity.

6. PLOS authors have the option to publish the peer review history of their article (what does this mean? ). If published, this will include your full peer review and any attached files.

**Do you want your identity to be public for this peer review?** For information about this choice, including consent withdrawal, please see our Privacy Policy .

Reviewer #1: No

Reviewer #2: No

Reviewer #3: **Yes: ** OMEED DARWEESH

---

## [Author Response · Author response to Decision Letter 1]

3 Dec 2024

We appreciate all reviewers for thoughtful suggestions and opinions. We improved manuscripts and responded to all the questions written below.

Introduction

The authors should briefly comment on the relationship between hypertension and COVID-19 and briefly describe the association between COVID-19 and outcomes such as AKI, the need for mechanical ventilation, and mortality.

Thank you for your insightful suggestion. We improved the introduction.

Line62

Moreover, not only diabetes but also older age, male, hypertension, obesity, cancer, and kidney failure were identified as the multiple risk factors of COVID-19 (3-5).

COVID-19 is associated with multi-organ damage, making the clinical manifestations critical for understanding the pathophysiology of the disease. It often leads to acute respiratory distress syndrome (ARDS), acute kidney injury (AKI) requiring hemodialysis, and other forms of organ failures(6). These complications contribute to increased admissions to intensive care unit (ICU), where the use of mechanical ventilation and dialysis is common(7).

Materials and Methods

What does it mean? The

-Diseases were indicated according to the International Classification of 83 Diseases, Tenth Revision (ICD-19) codes (6). The (????)-

Thank you for your comment. It was a mistake and corrected.

Line 91

Diseases, Tenth Revision (ICD-19) codes (6).

Study Population

How did you ensure that you evaluated patients diagnosed with diabetes mellitus? And

about COVID-19? Why didn't you run some blood tests? Was there blood glucose in the patients or glycated hemoglobin? What about creatinine, arterial blood gas analysis, and hemoglobin concentration?

This data is based on the combination data of diagnoses and payment system of medical insurance from each hospital. The diagnoses were based on the criteria of diabetes based on international standard and the medical doctors and hospitals claimed the insurance based on the diagnostic criteria of real clinical data with regular government inspections. The diagnosis of diabetes using DPC data has been reported to be accurate and we cited the article for the study of rationale. The data include the accurate diagnosis of diseases and payment information, but the blood data is not included unfortunately. We improved the explanation of the data in to make the article easy to understand.

Line 38

We collected inpatient medical data through Diagnosis procedure combination (DPC), the diagnoses and payment system of medical insurance, from each hospital.

Line 87

The DPC system is a classification system for hospital reimbursements from insurers with regular inspections by government in Japan.

Line 123

diagnosis of diabetes using DPC data has been reported to be accurate(11)

11. Kanehara R, Goto A, Goto M, Takahashi T, Iwasaki M, Noda M, et al. Validation Study of Diabetes Definitions Using Japanese Diagnosis Procedure Combination Data Among Hospitalized Patients. J Epidemiol. 2023;33(4):165-9.

Line 314

This study had some limitations. Our data provide information on diagnoses and treatments during hospitalization but not prior to hospitalization. Additionally, our data did not include information on blood examinations, vital signs, laboratory parameters, or radiographic data.

Why didn't you perform some matching between the diabetes and control groups, such as age, sex, or comorbidities? If you had, it would have made comparisons more robust.

Thank you for your insightful comments. As you pointed out, matching is a robust and powerful method for strengthening the causal interpretation of results when using sufficient confounders. We honestly analyzed the data using propensity score matching. However, for the following reasons, we discussed the data and decided to employ and choose multiple logistic regression to evaluate the statistical association between diabetes and COVID-related outcomes under adjustment through our internal discussions.

Firstly, our study aims to assess whether diabetes is a risk factor for several COVID-related outcomes, or in other words, whether it serves as a useful factor for predicting poor prognosis at the time of hospitalization. We believe that the multiple logistic regression we conducted is a straightforward and appropriate method for evaluating the association between diabetes and outcomes while adjusting for the key factors that were available in this study.

Secondly, even if all confounding factors could be collected, the choice between matching and regression is not straightforward, as each method has its advantages. For example, Brazauskas and Logan (2016, *1) highlight situations where regression may be superior to matching.

Lastly, this study is an observational study using real-world data (RWD), which offers the strength of evaluating the clinical significance of diabetes as a risk factor in actual clinical practice. However, as noted in the Discussion, certain critical factors, such as blood test results and vaccination history, were difficult to collect. This limitation can be considered a trade-off inherent in analyzing real-world clinical practice using RWD. Based on our findings, exploring the causal relationship between diabetes and the prognosis of COVID-19 patients through a more experimental study design is considered to be interesting for further research.

*1 Brazauskas R, Logan BR. Observational Studies: Matching or Regression? Biol Blood Marrow Transplant. 2016 Mar;22(3):557-63. doi: 10.1016/j.bbmt.2015.12.005. Epub 2015 Dec 19. PMID: 26712591; PMCID: PMC4756459.

Results

Would the authors be able to inform the frequency of patients with type 1 diabetes in the sample?

I added the information. The frequency of Type 1 diabetes is quite low in Japan (E Kawasaki, N Matsuura, K Eguchi. Type 1 diabetes in Japan: Review. Diabetologia. 2006;49:8). Also the therapy for Type 1 diabetes is centered to professional treatment hospitals which have departments of pediatric endocrinology therefore the frequency might be lower among this data.

Line 127

Among 3,097 cases of diabetes, those are mostly type 2 diabetes except 15 cases were type 1 diabetes (0.48%).

It would be interesting for you to build a binary logistic regression table using mortality as the response variable.

Thank you for your suggestion. We conducted a binary logistic regression analysis and included results in the article.

Line 210

We conducted a binary logistic regression analysis to evaluate the contribution of various factors to mortality of COVID-19. The analysis included age, BMI, sex (male), diabetes, and ambulance use as response variables. Among these, diabetes emerged as the most significant predictor of mortality (OR 2.67, p value <0.01) (Supplementary Table 2).

Supplementary Table 2. Multiple regression analysis of mortality adjusted with each factor, diabetes, age, sex (male), BMI, and ambulance.

You need to correct this sentence:

The data was extracted between February220 and November 2021

Thank you for your comment. It was a mistake and corrected.

Line 131

The data was extracted between February 2020 and November 2021.

The proportion of males was higher (54.9%) in the control group than in the diabetes group (68.2%) (odds ratio [OR] 1.75, 95% confidence interval [CI] 1.60–1.91). Report me about that, is 54.9% higher than 68,2%?

Thank you for your comment. It was a mistake and corrected.

Line138

The proportion of males was higher (68.2%) in the diabetes group than in the control group (54.9%) (odds ratio [OR] 1.75, 95% confidence interval [CI] 1.60–1.91, p-value<0.001).

For patients with 133 diabetes, the BMI was 25.6%, whereas that of the control was 23.6 (p <0.001). What does it mean? BMI, 25.6 kg/m²? And 23.6 kg/m²?

Thank you for your comment. It was a mistake and corrected.

Line140

For patients with diabetes, the BMI was 25.6 kg/m², whereas that of the control was 23.6 kg/m² (p <0.001)

What kind of hypoglycemic agents were used for patients with diabetes mellitus?

We created the table to show hypoglycemic agents for patients. We attached CSV file of breakdown of oral medications as supporting information.

Line 145

Among hypoglycemic agents, DPP-4 inhibitors were the most common medication (42.4%). All background of hypoglycemic agents is presented in data (Supplementary Table 1, supporting information).

Supplementary Table 1. Hypoglycemic agents for patients with diabetes. Abbreviations: AGI=Alpha-glucosidase inhibitor

Could the authors report the PaO2/FiO2 ratio at baseline in patients undergoing mechanical ventilation?

Blood data of PaO2/FiO2 is not included unfortunately. We added the information in limitations

Line 311

This study had some limitations. Our data provide information on diagnoses and treatments during hospitalization but not prior to hospitalization. Additionally, our data did not include information on blood examinations, vital signs, laboratory parameters, radiographic data including PaO2/FiO2 ratio and severity classification of COVID-19.

Discussion

The authors need to discuss the hyperglycemia observed in patients with infection or even the fact that infections, including SARS-CoV-2, decompensate diabetes mellitus.

Thank you for your suggestion. We added a paragraph with new citations in discussion.

Line 278

It is believed that COVID-19 infects not only the lungs but also kidneys and pancreatic beta cells, causing inflammation(14, 15). The viral infection depends on the expression of angiotensin-converting enzyme 2 (ACE2) and TMPRSS-2 transmembrane protease, which are the main cellular factors involved in viral entry. ACE2 is expressed in the lungs, and the virus infects the lungs and causes ARDS(16). COVID-19 stimulates strong immune response, resulting in a cytokine storm. It is possible that patients with diabetes are unable to suppress inflammation and that the virus enters the organs via those proteins, further exacerbating the inflammation and hyperglycemia(15, 17).

14. Khan S, Chen L, Yang CR, Raghuram V, Khundmiri SJ, Knepper MA. Does SARS-CoV-2 Infect the Kidney? J Am Soc Nephrol. 2020;31(12):2746-8.

15. Poloni TE, Moretti M, Medici V, Turturici E, Belli G, Cavriani E, et al. COVID-19 Pathology in the Lung, Kidney, Heart and Brain: The Different Roles of T-Cells, Macrophages, and Microthrombosis. Cells. 2022;11(19).

16. Zipeto D, Palmeira JDF, Argañaraz GA, Argañaraz ER. ACE2/ADAM17/TMPRSS2 Interplay May Be the Main Risk Factor for COVID-19. Front Immunol. 2020;11:576745.

17. Feldman EL, Savelieff MG, Hayek SS, Pennathur S, Kretzler M, Pop-Busui R. COVID-19 and Diabetes: A Collision and Collusion of Two Diseases. Diabetes. 2020;69(12):2549-65.

The authors need to discuss the relationship between the need for mechanical ventilation, AKI, dialysis requirement, and mortality in patients with diabetes mellitus and COVID-19.

Thank you for your suggestion. We added a paragraph with new citations in discussion.

Line289

In this study, diabetes was a significant factor in in-hospital mortality, mechanical ventilation, ICU stay, HD, and CHDF. AKI associated with COVID-19 is mainly caused by proximal tubular dysfunction. ACE2 expression is abundant in the proximal tubules of the kidney, and it is believed that SARS-CoV-2 infects these cells, causing damage(18, 19). AKI complications were reported in 36.6% of patients admitted with COVID-19(20). The combination of renal and respiratory failure is thought to have led to the use of mechanical ventilation, continuous hemodialysis (CHDF), and hemodialysis (HD) to support critically ill patients in the ICU.

18. Santoriello D, Khairallah P, Bomback AS, Xu K, Kudose S, Batal I, et al. Postmortem Kidney Pathology Findings in Patients with COVID-19. J Am Soc Nephrol. 2020;31(9):2158-67.

19. Werion A, Belkhir L, Perrot M, Schmit G, Aydin S, Chen Z, et al. SARS-CoV-2 causes a specific dysfunction of the kidney proximal tubule. Kidney Int. 2020;98(5):1296-307.

20. Silver SA, Beaubien-Souligny W, Shah PS, Harel S, Blum D, Kishibe T, et al. The Prevalence of Acute Kidney Injury in Patients Hospitalized With COVID-19 Infection: A Systematic Review and Meta-analysis. Kidney Med. 2021;3(1):83-98.e1.

Reviewer #2: I have some comments and questions for the authors

1. What is the reason for the existence of diabetes to augmented the probability of hospitalization via ambulance?

Thank you for your suggestion. We looked up the article but we couldn`t find the previous study in database such as pubmed. We assume that after infection of COVID-19, people stay at home while they observe the natural course of the disease for a while. Those with diabetes tend to have critically ill symptoms and they need to call ambulances finally. We`re running a retrospective study of the COVID-19 and diabetes in our hospital. We`re going to analyze the data from the point of view.

2. 77.8% of patients with diabetes used steroids for the treatment of pneumonia. I think it is important to know what percentage of pneumonias. What percentage were severe pneumonias?

Thank you for your question. I agree to know the percentage of pneumonias (or stages of COVID-19). But we do not have the information. We added it in limitation.

Line311

This study had some limitations. Our data provide information on diagnoses and treatments during hospitalization but not prior to hospitalization. Additionally, our data did not include information on blood examinations, vital signs, laboratory parameters, radiographic data including PaO2/FiO2 ratio and severity classification of COVID-19.

3. 7.8% of the control group and 17.9% of the group with diabetes required mechanical ventilation. What was the main reason why the patients required this support? Do you have respiratory data such as paO2/fiO2

We follow the medical guideline for COVID-19 in Japan. Basically, SpO2 is below 93% with mask with reservoir, or nasal high-flow, or positive pressure ventilation, we introduce mechanical ventilation for patients. We assume the patients could not maintain their oxygen level because of ARDS. We do not have the data of PaO2/FiO2. We updated the article.

Line71

Treatment of COVID-19 in Japan followed Japanese guidelines, including the introduction of mechanical ventilation(8).

8. Kato Y. Case Management of COVID-19 (Secondary Version). Jma j. 2021;4(3):191-7.

Line311

This study had some limitations. Our data provide information on diagnoses and treatments during hospitalization but not prior to hospitalization. Additionally, our data did not include information on blood examinations, vital signs, laboratory parameters, radiographic data including PaO2/FiO2 ratio and severity classification of COVID-19.

4. An important conclusion is that diabetes was an independent risk factor for in-hospital mortality. However, other conclusions such as risk factor for respirator use require further discussion.

Thank you for your suggestions. We improved discussion and added paragraphs with citations.

Line278

It is believed that COVID-19 infects not only the lungs but also kidneys and pancreatic beta cells, causing inflammation(14, 15). The viral infection depends on the expression of angiotensin-converting enzyme 2 (ACE2) and TMPRSS-2 transmembrane protease, which are the main cellular factors involved in viral entry. ACE2 is expressed in the lungs, and the virus infects the lungs and causes ARDS(16). COVID-19 stimulates strong immune response, resulting in a cytokine storm. It is possible that patients with diabetes are unable to suppress inflammation and that the virus enters the organs via those proteins, further exacerbating the inflammation and hyperglycemia(15, 17).

In this study, diabetes was a significant factor in in-hospital mortality, mechanical ventilation, ICU stay, HD, and CHDF. AKI associated with COVID-19 is mainly caused by proximal tubular dysfunction. ACE2 expression is abundant in the proximal tubules of the kidney, and it is believed that SARS-CoV-2 infects these cells, causing damage(18, 19). AKI complications were reported in 36.6% of patients admitted with COVID-19(20). The combination of renal and respiratory f

---

## [Decision Letter · Decision Letter 1]

9 Feb 2025

Diabetes with COVID-19 was a significant risk factor for mortality, mechanical ventilation, and renal replacement therapies: A multicenter retrospective study in Japan

PONE-D-24-22312R1

Dear Dr. Suwanai,

We’re pleased to inform you that your manuscript has been judged scientifically suitable for publication and will be formally accepted for publication once it meets all outstanding technical requirements.

Kind regards,

Fumihiro Yamaguchi

Academic Editor

PLOS ONE

Additional Editor Comments (optional):

Reviewers' comments:

Reviewer's Responses to Questions

**Comments to the Author**

1. If the authors have adequately addressed your comments raised in a previous round of review and you feel that this manuscript is now acceptable for publication, you may indicate that here to bypass the “Comments to the Author” section, enter your conflict of interest statement in the “Confidential to Editor” section, and submit your "Accept" recommendation.

Reviewer #1: All comments have been addressed

Reviewer #3: All comments have been addressed

2. Is the manuscript technically sound, and do the data support the conclusions?

Reviewer #1: Yes

Reviewer #3: Yes

3. Has the statistical analysis been performed appropriately and rigorously? 

Reviewer #1: Yes

Reviewer #3: Yes

4. Have the authors made all data underlying the findings in their manuscript fully available?

Reviewer #1: Yes

Reviewer #3: Yes

5. Is the manuscript presented in an intelligible fashion and written in standard English?

Reviewer #1: Yes

Reviewer #3: Yes

6. Review Comments to the Author

Reviewer #1: Dear Authors,

You have managed to do a multicenter retrospective analysis in 38 hospitals on the effect of COVID-19 on patients with diabetes mellitus.

I have no further questions.

Reviewer #3: (No Response)

7. PLOS authors have the option to publish the peer review history of their article (what does this mean? ). If published, this will include your full peer review and any attached files.

**Do you want your identity to be public for this peer review?** For information about this choice, including consent withdrawal, please see our Privacy Policy .

Reviewer #1: No

Reviewer #3: **Yes: ** Dr Omeed Omar Darweesh

---

## [Editor Report · Acceptance letter]

PONE-D-24-22312R1

PLOS ONE

Dear Dr. Suwanai,

I'm pleased to inform you that your manuscript has been deemed suitable for publication in PLOS ONE. Congratulations! Your manuscript is now being handed over to our production team.

Kind regards,

on behalf of

Dr. Fumihiro Yamaguchi

Academic Editor

PLOS ONE